8th ICML Workshop on Automated Machine Learning (2021)

# Latency-Aware Neural Architecture Search with Multi-Objective Bayesian Optimization

**David Eriksson**[*]                                                        DERIKSSON@FB.COM
*Facebook*

**Pierce I-Jen Chuang**[*]                                                   PICHUANG@FB.COM
*Facebook*

**Samuel Daulton**[*]                                                        SDAULTON@FB.COM
*Facebook*

**Peng Xia**                                                                PENGXIA@FB.COM
*Facebook*

**Akshat Shrivastava**                                                      AKSHATS@FB.COM
*Facebook*

**Arun Babu**                                                               ARBABU@FB.COM
*Facebook*

**Shicong Zhao**                                                            ZSC@FB.COM
*Facebook*

**Ahmed Aly**                                                               AHHEGAZY@FB.COM
*Facebook*

**Ganesh Venkatesh**                                                        GVEN@FB.COM
*Facebook*

**Maximilian Balandat**                                                     BALANDAT@FB.COM
*Facebook*

*∗ = Equal contribution.*

## Abstract

When tuning the architecture and hyperparameters of large machine learning models for on-device deployment, it is desirable to understand the optimal trade-offs between on-device latency and model accuracy. In this work, we leverage recent methodological advances in Bayesian optimization over high-dimensional search spaces and multi-objective Bayesian optimization to efficiently explore these trade-offs for a production-scale on-device natural language understanding model at Facebook.

## 1. Introduction

Neural architecture search (NAS) aims to provide an automated framework that identifies the optimal architecture for a deep neural network machine learning model given an evaluation criterion such as the model's predictive performance. The continuing trend towards deploying models on end user devices such as mobile phones has led to increased interest in optimizing multiple competing evaluation criteria to achieve an optimal balance between

predictive performance and computational complexity (e.g. total number of flops), memory footprint, and latency of the model. To address the NAS problem, the research community has developed a wide range of search algorithms, leveraging reinforcement learning (RL) (Zoph and Le, 2017; Tan et al., 2019), evolutionary search (ES) (Real et al., 2019; Liu et al., 2018), and weight-sharing (Cai et al., 2020; Yu et al., 2020; Wang et al., 2021), among others. However, RL and ES can incur prohibitively high computational costs because they require training and evaluating a large number of architectures. While weight sharing can achieve better sample complexity, it typically requires deeply integrating the optimization framework into the training and evaluation workflows, making it difficult to generalize to different production use-cases.

In this work, we aim to bridge this gap and provide a NAS methodology that requires *zero* code change to a user's training flow and can thus easily leverage existing large-scale training infrastructure while providing highly sample-efficient optimization of multiple competing objectives. We employ Bayesian optimization (BO), a popular method for black-box optimization of computationally expensive functions that achieves high sample-efficiency (Frazier, 2018). BO has been successfully used for tuning machine learning hyper-parameters for some time (Snoek et al., 2012; Turner et al., 2021), but only recently emerged as a promising approach for NAS (White et al., 2019; Falkner et al., 2018; Kandasamy et al., 2018; Shi et al., 2019; Parsa et al., 2020).

## 2. Use-case: On-Device Natural Language Understanding

We focus on tuning the architecture and hyperparameters of an on-device natural language understanding (NLU) model that is commonly used by conversational agents found in most mobile devices and smart speakers. The primary objective of the NLU model is to understand the user's semantic expression, and then to convert that expression into a structured decoupled representation that can be understood by downstream programs. As an example, a user may ask the conversational assistant "what is the weather in San Francisco?". In order for the assistant to reply back with an appropriate answer, e.g., "The weather in San Francisco is Sunny, with a high of 80 degrees", it needs to first understand the user's question. This is the primary objective of the NLU model, which converts the user's semantic expression into a representation that can be understood by downstream tasks. We adapt a structured semantic representation of utterance to accomplish this task. In this setting, the NLU model takes the aforementioned query as input and generates the following structured semantic representation, [IN: GET_WEATHER [SL: LOCATION San Francisco ] ], where IN and SL represent the *intent* and *slot*, respectively. The downstream program can then easily identify that the user is interested in the current weather in San Francisco.

The NLU model shown in Fig. 1 is an encoder-decoder non-autoregressive (NAR) architecture (Babu et al., 2021) based on the span pointer formulation (Shrivastava et al., 2021). The span pointer parser is a recently proposed semantic formulation that has shown to achieve state-of-the-art results on several task-oriented semantic parsing datasets (Shrivastava et al., 2021). An example of the span form representation is shown in Table 1.

The NLU model adapts a LightConv architecture consisting of a depth-wise convolution with weight sharing and a transformer-like multi-attention architecture to avoid recurrence. An input utterance of $n$ tokens (e.g., "what is the weather in San Francisco?" has a total

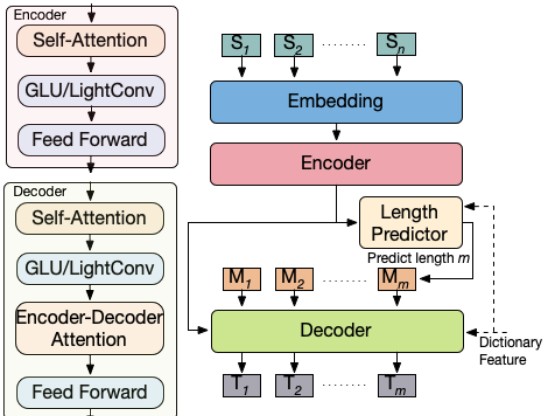

Figure 1: Non-Autoregressive Model Architecture of the NLU Semantic Parsing.

| Utterance | what is the weather in San Francisco |
| Index | 1  2  3     4    5   6     7 |
| --- | --- |
| Canonical Form | [IN: GET_WEATHER [SL: LOCATION  San Francisco ] ] |
| Span Form | [IN: GET_WEATHER [SL: LOCATION  6 7 ] ] |

Table 1: Comparison of the canonical and span forms of the decoupled frame representation. Given the utterance "what is the weather in San Francisco", our span form produces endpoints instead of text, reformulating the task from text generation to index prediction.

of 7 tokens) first go through the embedding and encoder layer to generate the contextual information. Then, a convolutional neural network (CNN) based length predictor takes this information and predicts the length of the final output token, labeled as $m$. For example, [IN: GET_WEATHER [SL: LOCATION San Francisco ] ] requires 8 output tokens for representation. On the decoder side, masked tokens with a length $m$ are initialized and propagated through the decoder layer to generate the final output tokens.

As the NLU model serves as the first stage in conversational assistants, it is crucial that it achieves high accuracy as the user experience largely depends on whether the users' semantic expression can be correctly translated. Conversational assistants operate over the user's language, potentially in privacy-sensitive situations such as when sending a message. For this reason, they generally run on the user's device ("on-device"), which comes at the cost of limited computational resources. While we generally expect a complex NLU model with a large number of parameters to achieve better accuracy, we also want short on-device inference time (latency) so as to provide a pleasant, responsive user experience. As complex NLU models with high accuracy also tend to have high latency, we are interested in exploring the trade-offs between these two objectives so that we can pick a model that offers an overall positive user experience by balancing quality and delivery speed of the suggestions.

## 3. Background

### 3.1 Multi-Objective Optimization

In multi-objective optimization (MOO), the goal is to maximize[1] a vector-valued objective $\boldsymbol{f}(\boldsymbol{x}) \in \mathbb{R}^M$ over a bounded set $\mathcal{X} \subset \mathbb{R}^d$. Typically, there is no single best solution. Rather, the goal is to identify the Pareto frontier: the set of optimal objective trade-offs such that improving one objective means degrading another. A point $\boldsymbol{f}(x)$ is *Pareto-optimal* if it is not *Pareto-dominated* by any other point. With knowledge of the Pareto-optimal trade-offs, a decision-maker can choose an objective trade-off according to their preferences. Typically, the goal in MOO is to identify a finite, approximate Pareto frontier within some fixed budget. The quality of a Pareto frontier is commonly assessed according to the *hypervolume* that is dominated by the Pareto frontier and bounded from below by a reference point that bounds the region of interest in objective-space and is usually supplied by the decision-maker.

### 3.2 Bayesian Optimization

Bayesian optimization (BO) is a sample-efficient methodology for optimizing expensive-to-evaluate black-box functions. BO leverages a probabilistic surrogate model, typically a Gaussian Process (GP) (Rasmussen, 2004). GPs are flexible non-parametric models that are specified by a mean function $\mu : \mathbb{R}^d \to \mathbb{R}$ and a (positive semi-definite) covariance function $k : \mathbb{R}^d \times \mathbb{R}^d \to \mathbb{R}$. In this work, we choose the mean function $\mu$ to be constant and the covariance function $k$ to be the popular Matérn-5/2 kernel. In addition to the mean constant, we also learn a signal variance $s^2$, a noise variance $\sigma^2$, and separate lengthscales $\ell_i$ for each input dimension. We learn the hyperparameters by optimizing the log-marginal likelihood, which has an analytic form (Rasmussen, 2004).

In addition, BO relies on an acquisition function that uses the GP model to provide a utility value for evaluating candidate points on the true black-box functions. In the MOO setting, common acquisition functions include the expected improvement (EI) with respect to a scalarized objective (Knowles, 2006), expected hypervolume improvement (EHVI) (Emmerich et al., 2006; Daulton et al., 2020), and information gain with respect to the Pareto frontier (Hernandez-Lobato et al., 2016; Suzuki et al., 2020).

## 4. Multi-Objective Bayesian NAS

We aim to explore the set of Pareto-optimal tradeoffs between model accuracy and latency over a *search space* containing parameters specifying the model architecture from Fig. 1.

### 4.1 Architecture Search Space

The model architecture hyper-parameters are given in Table 2 in Sec. B. For each of the main components (encoder, decoder, and the length predictor) in the NLU model, kernel_list determines the number of layers and the corresponding LightConv kernel size. As an example, a list of $[3, 5, 7]$ refers to 3 layers and that the first/second/third layer's LightConv kernel length is 3/5/7, respectively. Other important hyperparameters are (1) embed_dim and ffn_dim that determine the input and the transformer's feed-forward network (FFN)

---

1. We assume maximization without loss of generality.

width, respectively, and (2) attention_heads for the number of heads in the attention module. We encode each layer width as an integer parameter and use an additional parameter to control the number of layers included. This leads to a total of 24 parameters.

## 4.2 Fully Bayesian Inference with SAAS priors

Our 24-dimensional search space poses a challenge for standard GP models as described in Sec. 3.2. In Fig. 2 we show the leave-one-out cross-validation for two different model fitted to the two objectives for 100 different input configurations. We consider objective values relative to the performance of a base model (see Sec. 5 for more details). We compare a standard GP model with a maximum a posteriori (MAP) approach to the recently introduced sparse axis-aligned subspace (SAAS) prior (Eriksson and Jankowiak, 2021). The SAAS models places sparse priors on the inverse lengthscales in addition to a global shrinkage prior. When combined with the No-U-Turn sampler (NUTS) for inference, this leads to a model that picks out the most relevant lengthscales, making the model suitable for high-dimensional BO (see Sec. C for more details). As seen in Fig. 2, the SAAS prior provides a much better model fit for both objectives, making it a suitable modeling choice.

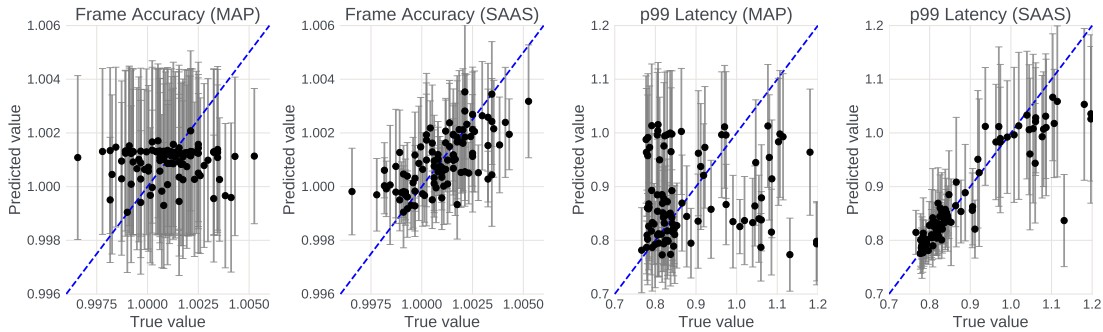

Figure 2: Leave-one-out cross-validation comparison for SAAS and MAP using 100 training configurations. Using the SAAS prior provides good fits for both objectives while MAP estimation is unable to provide accurate model fits.

## 4.3 Noisy Expected Hypervolume Improvement

We use the recently proposed parallel Noisy Expected Hypervolume Improvement acquisition function ($q$NEHVI) (Daulton et al., 2021), which has been shown to perform well with high levels of parallelism and under noisy observations. As we use fully Bayesian inference, we integrate the acquisition function over the posterior distribution $p(\psi|\mathcal{D})$ of the GP hyperparameters $\psi$ given the observed data $\mathcal{D}$:

$$\alpha_{q\text{NEHVI-MCMC}}(\mathcal{X}_{\text{cand}}) = \int_{\psi} \alpha_{q\text{NEHVI}}(\mathcal{X}_{\text{cand}}|\psi)p(\psi|\mathcal{D})d\psi \tag{1}$$

where $X_{\text{cand}}$ denotes set of $q$ new candidates $X_{\text{cand}} = \{\boldsymbol{x}_1, ..., \boldsymbol{x}_q\}$. Since the integral in (1) is intractable, we approximate the integral using $N_{\text{MCMC}}$ Monte Carlo (MC) samples (see Sec. D for additional details).

## 5. Experimental results

The goal is to maximize the accuracy of the model described in Sec. 2 on a held-out evaluation set while also minimize the on-device p99 latency (the 99th percentile of the latency distribution). Latency here defined as the time between when the input utterance becomes available and the model generates the final structured semantic representation. To produce a stable estimate of the p99 latency, a tested model is evaluated repeatedly many times on the same device. We select a reference point to bound the area of interest based on an existing model with hyperparameters selected using domain knowledge. We optimize the objectives *relative* to this reference point, which results in a reference point of $(1, 1)$.

We consider a computational budget of 240 evaluations and launch function evaluations asynchronously with parallelism of $q = 16$. For BO, we use 32 initial points from a scrambled Sobol sequence. To do inference in the SAAS model we rely on the open-source implementation of NUTS in Pyro (Bingham et al., 2019). We use $q$NEHVI as implemented in BoTorch (Balandat et al., 2020). The results from using BO as well as Sobol search are shown in Fig. 3. Sobol was only able to find a single configuration that outperformed the reference point. On the other hand, our BO method was able to explore the trade-offs and improve the p99 latency by more than 25% while also improving model accuracy.

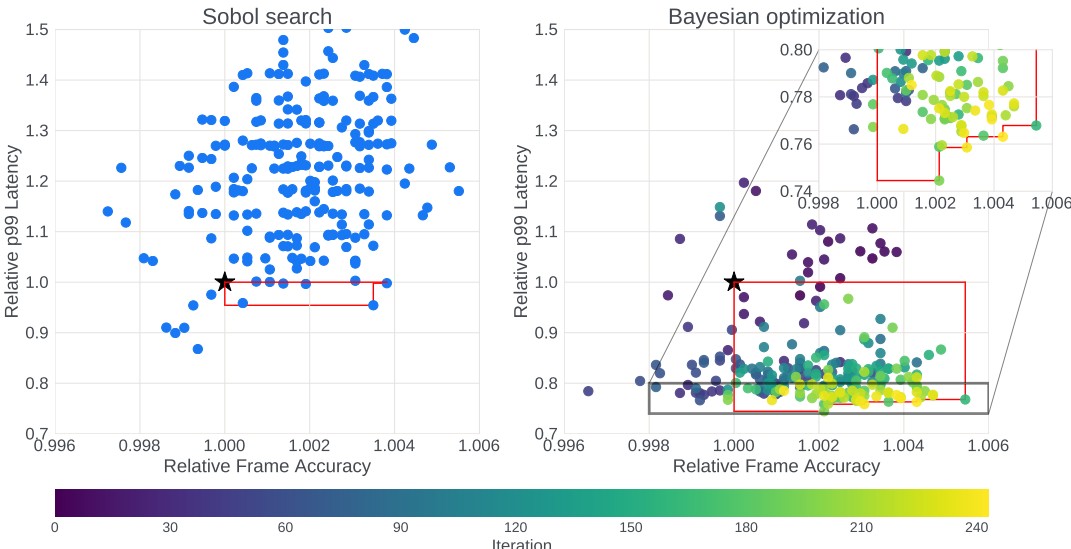

Figure 3: (Left) Sobol search is only able to find two points that improve upon the reference point. (Right) BO is able to successfully explore the trade-off between latency and accuracy.

## 6. Conclusion

We introduced a new BO method for sample-efficient multi-objective NAS. Our approach combines the SAAS prior for high-dimensional BO (Eriksson and Jankowiak, 2021) with the $q$NEHVI acquisition function (Daulton et al., 2021). When applied to a production-scale on-device natural language understanding model, our method was able to successfully explore the trade-off between model accuracy and on-device latency.

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

## Appendix A. Example json

We rely on PyText, which is a deep-learning based language modeling framework built on PyTorch, and is the primary framework for a variety of production NLP models at Facebook, including semantic parsing. PyText provides a flexible way to generate the deep learning model through a json configuration file, thus providing an easy way to integrate with BO without major code refactoring. In this simplified example, a 4-layer encoder, 1-layer decoder transformer-like model is generated. The encoder's embedding and feed-forward dimension are 128 and 192, respectively. At single-layer decoder end, the decoder's lightweight convolution's kernel size is 12, uses GLU for activation, and the attention module is a multi-headed configuration with the number of heads equals to two. This flexible expression allows the back-end NAS engine to be seamlessly integrated with the model training framework.

```json
{
    "encoder": {
        "encoder_embed_dim": 128,
        "encoder_ffn_embed_dim": 192,
        "encoder_kernel_list": [3, 5, 5, 7]
    },
    "decoder": {
        "decoder_embed_dim": 156,
        "decoder_output_dim": 128,
        "decoder_glu": true,
        "self_attention_heads": 2,
        "decoder_kernel_size_list": [12]
    }
    // config continues
}
```

Listing 1: Example of transformer-like model json configuration. This type of json configuration is the default in the PyText framework, and consequently any PyText machine learning engineer will be familiar with this format.

## Appendix B. Search space

A summary of the tunable parameters is given in Table 2. The kernel sizes are represented by one integer parameter for each layer and then one additional integer parameter that controls the length. In particular, we always optimize the acquisition function over all widths as well as the number of layers, but only include the first num_layers width in the resulting model. We emphasize that the representation of the kernel sizes could be further improved by taking into account the hierarchical structure, e.g., we do not need a parameter for the width of the 6th layer if we condition on the encoder length being 5. This results

in 7 parameters for the encoder, 3 for the decoder, and 3 for the convolution. In addition, there are 10 additional integer parameters and 1 boolean parameter, resulting in a total of 24 parameters.

| Parameter | Default | Search Space | Description |
|---|---|---|---|
| **Encoder** | | | |
| kernel_list | [3, 3, 5, 9, 7] | [3, 3, 3, 3], ... | list of length 4-6, drawn from [3, 5, 7, 9] |
| embed_dim | 128 | 128, 136, ..., 192 | input dimension |
| self_attention | 2 | 1, 2, 4 | number of self-attention head |
| ffn_dim | 40 | 32, 40, ..., 192 | feed-forward network (FFN) width |
| normalized | True | True, False | apply normalization before the FFN |
| **Decoder** | | | |
| kernel_list | [13, 9] | [7, 7], [7, 9], ... | list of length 1-2, drawn from [7, 9, 11, 13, 15] |
| self_attention | 1 | 1, 2, 4 | number of self-attention head |
| attention_heads | 2 | 1, 2, 4 | number of cross-attention head |
| ffn_dim | 144 | 128, 144, ..., 512 | FFN width |
| **Length Predictor** | | | |
| kernel_list | [3, 7] | [3], [3, 5], ... | list of length 1-2, drawn from [3, 5, 7] |
| dim | 192 | 32, 40, ..., 192 | convolution width |
| num_head | 4 | 1, 2, 4 | number of attention head |
| **Embedding** | | | |
| char_embed_dim | 8 | 8, 12, ..., 24 | character embedding dimension |
| proj_dim | 12 | 8, 12, ..., 24 | last layer projection dimension |

Table 2: List of the tunable parameters (i.e., search space) of the NLU model

## Appendix C. GP fitting

Before fitting the GP model we standardize the output values for each objective to have zero mean and unit variance. We also linearly scale the inputs to lie in the domain $[0, 1]^d$. Recall from Sec. 3.2 that we use a separate lengthscale $\ell_i$ for each input dimension. Following Eriksson and Jankowiak (2021), we use a global shrinkage $\tau \sim \mathcal{HC}(0.1)$ and priors $1/\ell_i \sim \mathcal{HC}(\tau)$, where $\mathcal{HC}$ is the half-Cauchy distribution. As all latent variables in the SAAS model are continuous, we can use the No-U-Turn sampler (NUTS) for inference, integrating out $f$ analytically in the log-marginal likelihood formulation. The global shrinkage is controlled via $\tau$ and its values will naturally concentrate around zero because of the $\mathcal{HC}$ prior. As the inverse lengthscales are also governed by a $\mathcal{HC}$ prior, they will also concentrate around zero and the resulting model will, in absence of strong contrary evidence from the data, have large lengthscales and thus "turn off" the majority of dimensions.

In addition to the lengthscale priors we use a $\Gamma(0.9, 10)$ prior for the noise variance, a $\Gamma(2.0, 0.15)$ prior for the signal variance, and a $U[-1, 1]$ prior for the constant mean. For NUTS, we use 512 warmup steps before collecting a total of 256 samples. Finally, we apply thinning and keep only every 16th sample, leaving us with a total of 32 hyperparam-

eter samples to average over when computing the acquisition function for any given set of candidate points.

## Appendix D. Noisy Expected Hypervolume Improvement

Snoek et al. (2012) proposed a similar fully Bayesian treatment of EI, but to our knowledge, no previous work on multi-objective optimization has considered integrating an EHVI-based acquisition function over the posterior distribution of the GP hyperparameters. Since the integral in (1) is intractable, we approximate it using $N_{\text{MCMC}}$ Monte Carlo (MC) samples:

$$\hat{\alpha}_{q\text{NEHVI-MCMC}}(\mathcal{X}_{\text{cand}}) = \frac{1}{N_{\text{MCMC}}} \sum_{n=1}^{N_{\text{MCMC}}} \alpha_{q\text{NEHVI}}(\mathcal{X}_{\text{cand}}|\psi_n)$$

$\alpha_{q\text{NEHVI}}$ also includes an intractable integral and therefore is itself approximated with $N$ MC samples. We integrate over the posterior distribution of the objectives at pending points to account for the previously selected candidates that are currently being evaluated. The cached box decomposition approach (CBD) used in qNEHVI can be used to efficiently compute $\hat{\alpha}_{q\text{NEHVI-MCMC}}$ by caching $N_{\text{MCMC}}N$ box decompositions. See Daulton et al. (2021) for details on efficient computation using CBD.

## Appendix E. Hypervolume improvement

Fig. 4 illustrates that BO achieves a much larger hypervolume compared to Sobol with respect to the reference point $(1, 1)$. BO immediately makes progress after the initial 32 Sobol points and consistently makes progress as the GP model becomes more accurate.

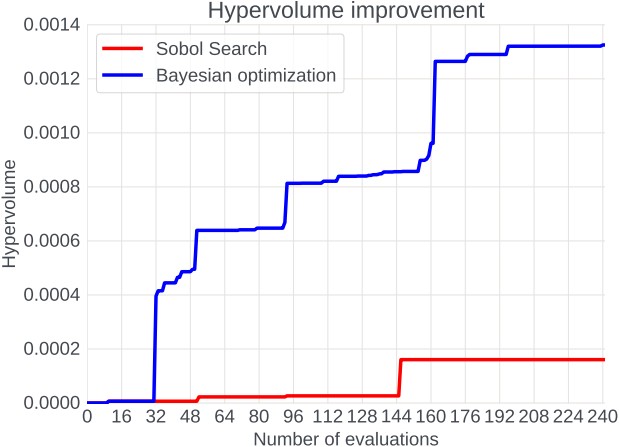

Figure 4: We see that BO improves the hypervolume quickly after the initial Sobol batch and makes continuous improvement until the evaluation budget is exhausted.

