# OpenReview forum: "Latency-Aware Neural Architecture Search with Multi-Objective Bayesian Optimization"
_ICML.cc/2021/Workshop/AutoML — AutoML@ICML2021 Poster_

### Official Review · Reviewer_rSFa · 2021-06-04
**Nice Application, Incremental Technical Contribution, Limited Experiment**

**Rating:** 6
**Confidence:** 4

**Review:**

The paper presents a Bayesian optimization approach to optimize the architecture and hyperparameters of a ML model for natural language understanding. A focus of this paper is to perform multi-objective optimization to have high accuracy and high latency property for on-device deployment.

Strengths:
Interesting applications for on-device deployment.
Well written and presentation.
The combination of high-dim optimization and multi-objective BO is relatively new.

Weaknesses:
The technical contribution is limited. The proposed method is a straightforward combination of existing algorithms.
The motivation behind the chosen algorithm is unclear, e.g., can we replace Noisy Expected Hypervolume Improvement with something else? or what happens if we dont use SAAS prior?
The experiment is limited. We expect to have comparisons with other common approaches for multi-objective and high-dim BO.

---

### Official Review · Reviewer_HV4q · 2021-06-11
**No novelty, but a nice practical application of already existing techniques.**

**Rating:** 7
**Confidence:** 4

**Review:**


This paper proposes the use of Bayesian optimization for finding a good configuration of a deep neural network related to a natural language understanding model. The task addressed has two objectives and requires the use of multi-objective Bayesian optimization methods. The objectives considered are the latency and the accuracy, which are conflictive. A particular prior for the GP models is considered, which is expected to work better on high-dimensional sub-spaces. The parallel noisy expected hyper-volume improvement acquisition function is used in the BO algorithm. The BO approach is compared with a uniform exploration of the input space using a Sobol sequence. The results show that BO methods obtain better configurations that trade-off the two objectives.

Overall I think that this paper is a nice implementation of already known methods to solve a challenging problem. I think that the paper is interesting and, while it has not that much novelty on it, it will attract the attention of the  workshop attendees. It is also a relevant paper for the workshop.

---

### Official Review · Reviewer_B7v1 · 2021-06-13
**Reasonable combination of two existing techniques**

**Rating:** 6
**Confidence:** 5

**Review:**

The authors address multi-objective NAS with Bayesian optimization. For this purpose they combine two recently introduced techniques. A GP with sparse axis-aligned subspace prior is used to meet the problem introduced by high dimensional search spaces. The Noisy Expected Hypervolume Improvement acquisition function is used to build a Pareto front. This system is then used to optimize the hyperparameters of an NLU model with respect to accuracy and latency.

The paper is well-written and the use-case and the design choices for the system are well-motivated. Results show clear improvements over a simple baseline.
I have few remarks how the work can be further improved:
i) The authors could point out the limitations of the method, i.e. that it requires a fixed length vector to represent the neural network. Thus, the method would not generalize to NAS (where you usually also adapt the computation graph and not only few hyperparameters).
ii) A section discussing general work on multi-objective NAS could be added.
iii) A stronger baseline could be added.

Smaller comments are:

i) Half of the references given for the claim that BO "emerged as a promising approach for NAS" do not use BO.
ii) The authors claim to present the first BO method for the multi-objective BO setting. However, e.g. Shi et al (cited by the authors) already do this. Another work is
Parsa et al: Bayesian Multi-objective Hyperparameter Optimization for Accurate, Fast, and Efficient Neural Network Accelerator Design

Typo: Section 5: "...using domain knowledgeWhat . We..."

---

### Decision · Program_Chairs · 2021-06-21

Accept (Poster)